# IMPROVING LEARNING TO BRANCH VIA REINFORCEMENT LEARNING

## ABSTRACT

Branch-and-Bound (B&B) is a general and widely used algorithm paradigm for solving Mixed Integer Programming (MIP). Recently there is a surge of interest in designing learning-based branching policies as a fast approximation of strong branching, a human-designed heuristic. In this work, we argue that strong branching is not a good expert to imitate for its poor decision quality when turning off its side effects in solving branch linear programming. To obtain more effective and non-myopic policies than a local heuristic, we formulate the branching process in MIP as reinforcement learning (RL) and design a novel set representation and distance function for the B&B process associated with a policy. Based on such representation, we develop a novelty search evolutionary strategy for optimizing the policy. Across a range of NP-hard problems, our trained RL agent significantly outperforms expert-designed branching rules and the state-of-the-art learning-based branching methods in terms of both speed and effectiveness. Our results suggest that with carefully designed policy networks and learning algorithms, reinforcement learning has the potential to advance algorithms for solving MIPs.

## 1 INTRODUCTION

Mixed Integer Programming (MIP) has been applied widely in many real-world problems, such as scheduling (Barnhart et al., 2003) and transportation (Melo & Wolsey, 2012). Branch and Bound (B&B) is a general and widely used paradigm for solving MIP problems (Wolsey & Nemhauser, 1999). B&B recursively partitions the solution space into a search tree and compute relaxation bounds along the way to prune subtrees that provably can not contain an optimal solution. This iterative process requires sequential decision makings: *node selection*: selecting the next solution space to evaluate, *variable selection*: selecting the variable by which to partition the solution space (Achterberg & Berthold, 2009). In this work, we focus on learning a variable selection strategy, which is the core of the B&B algorithm (Achterberg & Wunderling, 2013).

Very often, instances from the same MIP problem family are solved repeatedly in industry, which gives rise to the opportunity for learning to improve the variable selection policy (Bengio et al., 2020). Based on the human-designed heuristics, Di Liberto et al. (2016) learn a classifier that dynamically selects an existing rule to perform variable selection; Balcan et al. (2018) consider a weighted score of multiple heuristics and analyse the sample complexity of finding such a good weight. The first step towards learning a variable selection policy was taken by Khalil et al. (2016), who learn an instance customized policy in an online fashion, as well as Alvarez et al. (2017) and Hansknecht et al. (2018) who learn a branching rule offline on a collection of similar instances. Those methods need extensively feature engineering and require strong domain knowledge in MIP. To avoid that, Gasse et al. (2019) propose a graph convolutional neural network approach to obtain competitive performance, only requiring raw features provided by the solver. In each case, the branching policy is learned by imitating the decision of strong branching as it consistently leads to the smallest B&B trees empirically (Achterberg et al., 2005).

In this work, we argue that strong branching is not a good expert to imitate. The excellent performance (the smallest B&B tree) of strong branching relies mostly on the information obtained in solving branch linear programming (LP) rather than the decision it makes. This factor prevents learning a good policy by imitating only the decision made by strong branching. To obtain more effective and non-myopic policies, i.e. minimizing the total solving nodes rather than maximizing the immediate duality gap gap, we use reinforcement learning (RL) and model the variable selection process as a Markov Decision Process (MDP). Though the MDP formulation for MIP has been mentioned in the previous works (Gasse et al., 2019; Etheve et al., 2020), the advantage of RL has not been demonstrated clearly in literature.

The challenges of using RL are multi-fold. First, the state space is a complex search tree, which can involve hundreds or thousands of nodes (with a linear program on each node) and evolve over time. In the meanwhile, the objective of MIP is to solve problems faster. Hence a trade-off between decision quality and computation time is required when representing the state and designing a policy based on this state representation. Second, learning a branching policy by RL requires rolling out on a distribution of instances. Moreover, for each instance, the solving trajectory could contain thousands of steps and actions can have long-lasting effects. These result in a large variance in gradient estimation. Third, each step of variable selection can have hundreds of candidates. The large action set makes the exploration in MIP very hard.

In this work, we address these challenges by designing a policy network inspired by primal-dual iteration and employing a novelty search evolutionary strategy (NS-ES) to improve the policy. For efficiency-effectiveness trade-off, the primal-dual policy ignores the redundant information and makes high-quality decisions on the fly. For reducing variance, the ES algorithm is an attractive choice as its gradient estimation is independent of the trajectory length (Salimans et al., 2017). For exploration, we introduce a new representation of the B&B solving process employed by novelty search (Conti et al., 2018) to encourage visiting new states.

We evaluate our RL trained agent over a range of problems (namely, set covering, maximum independent set, capacitated facility location). The experiments show that our approach significantly outperforms state-of-the-art human-designed heuristics (Achterberg & Berthold, 2009) as well as imitation based learning methods (Khalil et al., 2016; Gasse et al., 2019). In the ablation study, we compare our primal-dual policy net with GCN (Gasse et al., 2019), our novelty based ES with vanilla ES (Salimans et al., 2017). The results confirm that both our policy network and the novelty search evolutionary strategy are indispensable for the success of the RL agent. In summary, our main contributions are the followings:

- We point out the overestimation of the decision quality of strong branching and suggest that methods other than imitating strong branching are needed to find better variable selection policy.
- We model the variable selection process as MDP and design a novel policy net based on primal-dual iteration over reduced LP relaxation.
- We introduce a novel set representation and optimal transport distance for the branching process associated with a policy, based on which we train our RL agent using novelty search evolution strategy and obtain substantial improvements in empirical evaluation.

## 2 BACKGROUND

**Mixed Integer Programming**. MIP is an optimization problem, which is typically formulated as

$$\min_{\mathbf{x} \in \mathbb{R}^n} \{\mathbf{c}^T \mathbf{x} : A\mathbf{x} \leq \boldsymbol{b}, \boldsymbol{\ell} \leq \mathbf{x} \leq \boldsymbol{u}, x_j \in \mathbb{Z}, \forall j \in J\} \tag{1}$$

where $\mathbf{c} \in \mathbb{R}^n$ is the objective vector, $A \in \mathbb{R}^{m \times n}$ is the constraint coefficient matrix, $\boldsymbol{b} \in \mathbb{R}^m$ is the constraint vector, $\boldsymbol{\ell}, \boldsymbol{u} \in \mathbb{R}^n$ are the variable bounds. The set $J \subseteq \{1, \cdots, n\}$ is an index set for integer variables. We denote the feasible region of $x$ as $\mathcal{X}$.

**Linear Programming Relaxation**. LP relaxation is an important building block for solving MIP problems, where the integer constraints are removed:

$$\min_{\mathbf{x}\in\mathbb{R}^n}\ \{\mathbf{c}^T\mathbf{x} : A\mathbf{x} \leq \boldsymbol{b}, \boldsymbol{\ell} \leq \mathbf{x} \leq \boldsymbol{u}\}. \tag{2}$$

**Branch and Bound**. LP based B&B is the most successful method in solving MIP. A typical LP based B&B algorithm for solving MIP looks as Algorithm 1 (Achterberg et al., 2005).

It consists of two major decisions: *node selection*, in line 3, and *variable selection*, in line 7. In this paper, we will focus on the *variable selection*. Given a LP relaxation and its optimal solution $\hat{x}$, the *variable selection* means selecting an index $j$. Then, branching splits the current problem into two subproblems, each representing the original LP relaxation with a new constraint $x_j \leq \lfloor \hat{x}_j \rfloor$ for $Q_j^-$ and $x_j \geq \lceil \hat{x}_j \rceil$ for $Q_j^+$ respectively. This procedure can be visualized by a binary tree, which is commonly called search tree. We give a simple visualization in Section A.1.

---

**Algorithm 1:** Branch and Bound

**Input:** A MIP $P$ in form Equation 1
**Output:** An optimal solution set $x^*$ and optimal value $c^*$

1 Initialize the problem set $S := \{P_{LP}\}$. where $P_{LP}$ is in form Equation 2. Set $x^* = \phi, c^* = \infty$ ;
2 If $S = \phi$, exit by returning $x^*$ and $c^*$ ;
3 Select and pop a LP relaxation $Q \in S$ ;
4 Solve $Q$ with optimal solution $\hat{x}$ and optimal value $\hat{c}$ ;
5 If $\hat{c} \geq c^*$, go to 2 ;
6 If $\hat{x} \in \mathcal{X}$, set $x^* = \hat{x}$, $c^* = \hat{c}$, go to 2 ;
7 Select variable $j$, split $Q$ into two subproblems $Q_j^+$ and $Q_j^-$, add them to $S$ and go to 3 ;

---

**Evolution Strategy**. Evolution Strategies (ES) is a class of black box optimization algorithm (Rechenberg, 1978). In this work, we refer to the definition in Natural Evolution Strategies (NES) (Wierstra et al., 2008). NES represents the population as a distribution of parameter vectors $\theta$ characterized by parameters $\phi : p_\phi(\theta)$. NES optimizes $\phi$ to maximize the expectation of a fitness $f(\theta)$ over the population $\mathbb{E}_{\theta \sim p_\phi}[f(\theta)]$. In recent work, Salimans et al. (2017) outlines a version of NES applied to standard RL benchmark problems, where $\theta$ parameterizes the policy $\pi_\theta$, $\phi_t = (\theta_t, \sigma)$ parameterizes a Gaussian distribution $p_\phi(\theta) = \mathcal{N}(\theta_t, \sigma^2 I)$ and $f(\theta)$ is the cumulative reward $R(\theta)$ over a full agent interaction. At every iteration, Salimans et al. (2017) apply $n$ additive Gaussian noises to the current parameter and update the population as

$$\theta_{t+1} = \theta_t + \alpha \frac{1}{n\sigma} \sum_{i=1}^{n} f(\theta_t + \sigma\epsilon_i)\epsilon_i \tag{3}$$

To encourage exploration, Conti et al. (2018) propose Novelty Search Evolution Strategy (NS-ES). In NS-ES, the fitness function $f(\theta) = \lambda N(\theta) + (1-\lambda)R(\theta)$ is selected as a combination of domain specific novelty score $N$ and cumulative reward $R$, where $\lambda$ is the balancing weight.

## 3 WHY IMITATING STRONG BRANCHING IS NOT GOOD

Strong branching is a human-designed heuristic, which solves all possible branch LPs $Q_j^+, Q_j^-$ ahead of branching. As strong branching usually produces the smallest B&B search trees (Achterberg, 2009), many learning-based variable selection policy are trained by mimicking strong branching (Gasse et al., 2019; Khalil et al., 2016; Alvarez et al., 2017; Hansknecht et al., 2018). However, we claim that strong branching is not a good expert: the reason strong branching can produce a small search tree is the reduction obtained in solving branch LP, rather than its decision quality. Specifically, (i) Strong branching can check lines 5, 6 in Algorithm 1 before branching. If the pruning condition is satisfied, strong branching does not need to add the subproblem into the problem set $S$. (ii) Strong branching can strengthen other LP relaxations in the problem set $S$ via domain propagation (Rodosek et al., 1999) and conflict analysis (Achterberg, 2007). For example, if strong branching finds $x_1 \geq 1$ and $x_2 \geq 1$ can be pruned during solving branch LP, then any other LP relaxations containing $x_1 \geq 1$ can be strengthened by adding $x_2 \leq 0$. These two reductions are

the direct consequence of solving branch LP, and they can not be learned by a variable selection policy. (iii) Strong branching activates primal heuristics (Berthold, 2006) after solving LPs.

To examine the decision quality of strong branching, we employ vanilla full strong branching (Gamrath et al., 2020), which takes the same decision as full strong branching, while the side-effect of solving branch LP is switched off. Experiments in Section 5.2 show that vanilla full strong branching has poor decision quality. Hence, imitating strong branching is not a wise choice for learning variable selection policy.

## 4 METHOD

Due to line 5 in Algorithm 1, a good variable selection policy can significantly improve solving efficiency. To illustrate how to improve variable selection policy, we organize this section in three parts. First, we present our formulation of the variable selection process as a RL problem. Next, we introduce the LP relaxation based state representation and the primal-dual based policy network. Then, we introduce our branching process representation and the corresponding NS-ES training algorithm.

### 4.1 RL FORMULATION

Let the B&B algorithm and problem distribution $\mathcal{D}$ be the environment. The sequential decision making of variable selection can be formulated as a Markov decision process. We specify state space $\mathcal{S}$, action space $\mathcal{A}$, transition $\mathcal{P}$ and reward $r$ as follows

- **State Space**. At iteration $t$, node selection policy will pop out a LP relaxation $P_{LP}$ from the problem set $S$. We set the representation of the state to $s_t = \{P_{LP}, J, S\}$, where $J$ is the index set of integer variables.
- **Action Space**. At iteration $t$, the action space is the index set of non-fixed integer variables determined by the relaxation: $\mathcal{A}(s_t) = \{j \in J : \ell_j < u_j\}$.
- **Transition**. Given state $s_t$ and action $a_t$, the new state is determined by the node selection policy.
- **Reward**. As our target is solving the problem faster, we set the reward $r_t = -1$ with discount $\gamma = 1$. Maximizing the cumulative reward encourages the agent solving problems with less steps.

In commercial solver, the solving process is much more complicated than the B&B stated in Algorithm 1. For example, between line 3 and line 4, primal heuristics could be used to detect feasible solutions, and cutting planes could be applied to strengthen the LP relaxation. These components in solver introduce more randomness in transition, but our formulation is still valid.

### 4.2 PRIMAL DUAL POLICY NET

**Reduced LP**. In the solving process, the variable bounds keep changing due to branching. Thus, we obtain our reduced LP relaxation by the following two steps: 1) remove fixed variables $x_j$, where $\ell_j = u_j$, and plug their values into the constraints; 2) remove trivial constraints, where $\max_{\ell \leq \mathbf{x} \leq \mathbf{u}} \sum_j A_{ij} x_j \leq b_i$. In the view of primal-dual iteration, the LP relaxation has Lagrangian form:

$$\min_{\mathbf{x}} \max_{\boldsymbol{\lambda}} \mathbf{c}^T \mathbf{x} + \boldsymbol{\lambda}^T (A\mathbf{x} - \boldsymbol{b}), \quad \text{s.t.} \quad \boldsymbol{\ell} \leq \mathbf{x} \leq \boldsymbol{u}, \mathbf{0} \leq \boldsymbol{\lambda} \tag{4}$$

where variables and constraints naturally form a bipartite graph. In the primal-dual iteration over Equation 4, fixed variables and trivial constraints always pass zero and have no interaction with other variables.

**PD policy net**. We parameterize our policy network $\pi_\theta(a_t|s_t)$ as a primal-dual iteration over the reduced LP relaxation by message passing

$$\mathbf{Y}_i \leftarrow \mathbf{f}_\mathcal{C}\Big(\mathbf{Y}_i, \sum_j \mathbf{A}_{ij} \, \mathbf{m}_\mathcal{C}\left(\mathbf{X}_j\right)\Big), \quad \mathbf{X}_j \leftarrow \mathbf{f}_\mathcal{V}\Big(\mathbf{X}_j, \sum_i \mathbf{A}_{ij} \, \mathbf{m}_\mathcal{V}\left(\mathbf{Y}_i\right)\Big) \tag{5}$$

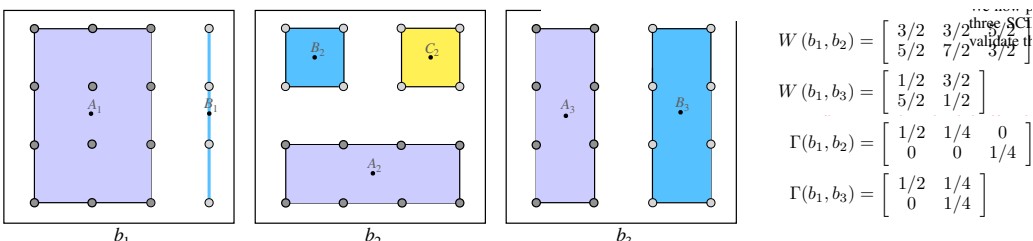

Figure 1: (left) three policies $\pi_1$, $\pi_2$ and $\pi_3$ produce three sets of polytopes $b_1$, $b_2$ and $b_3$ respectively for the same problem $Q$, (right) example cost matrix $W$ and transportation matrix $\Gamma$.

where $f_{\mathcal{C}}, f_{\mathcal{V}}$ are two-hidden-layers neural networks, $m_{\mathcal{C}}, m_{\mathcal{V}}$ are one hidden layer neural networks, $A_{ij}$ is the entry in the reduced constraint matrix $A$ and $\mathbf{X}, \mathbf{Y}$ are the embedding for variables and constraints initialized by $P_{LP}$ and $J$. As mentioned above, the original primal-dual iterations only occurs on the reduced LP hence, our message passing in Equation 5 is defined only on the reduced graph. For efficiency, we do not include problem set $S$, which makes it a partial observable MDP (Astrom, 1965). After two iterations of Equation 5, the variable embedding $\mathbf{X}$ is passed to a two-hidden-layer neural network score function $f_{\mathcal{S}}$ and the output is the final score for each variable. Since the state reduction and message passing are both inspired by primal-dual iteration, we call it PD policy. A more detailed discussion and comparison with GCN (Gasse et al., 2019) can be found at section A.2.2.

### 4.3 SET REPRESENTATION FOR POLICY AND OPTIMAL TRANSPORT DISTANCE

We train the RL agent using evolution strategy similar to NSR-ES (Conti et al., 2018) and we need to define the novelty score for B&B process. In the general B&B algorithm, the solving process can be represented by a search tree, where each leaf is a solved subproblem. Given a branch policy $\pi$ and an instance $Q$, we define our representation $b(\pi, Q) = \{R_1, \cdots, R_H\}$ as the collection of leaf subproblems on the complete search tree . Focusing on MIP, a subproblem $R_i$ is a LP relaxation which can be represented by its feasible region, a polytope. For example, in Figure 1, $b_1$, $b_2$ and $b_3$ are the set of polytopes produced by three different policies $\pi_1$, $\pi_2$ and $\pi_3$ respectively. And $b_1 = \{A_1, B_1\}$ is a set of two polytopes (leaf subproblems), $b_2 = \{A_2, B_2, C_2\}$ is a set of three polytopes, and $b_3 = \{A_3, B_3\}$ is a set of two polytopes. For computational efficiency, we ignore the constraints and only consider variable bounds such that every polytope is a box.

For each polytope $R_i$ (leaf subproblem), we define the weight function $w(\cdot)$ and distance function $d(\cdot, \cdot)$ between two polytopes $R_i$ and $R_j$ as

- $w(R_i) := \#\{x \in R_i : x \text{ is a feasible solution for } Q\}$.
- $d(R_i, R_j) := \|g_i - g_j\|_1$, where $g_i$ and $g_j$ are the center of mass for $R_i$ and $R_j$ respectively.

For example, in Figure 1, we have $w(A_1) = 12, d(A_1, A_2) = \frac{3}{2}$. Then we can map the representation $b = \{R_1, \cdots, R_H\}$ to a simplex $p(b) \in \Delta^{H-1}$ by normalizing the weights $p(R_j) = w(R_j)/\sum_{i=1}^{H} w(R_i)$, and compute a cost matrix $W_{ij} = d(R_i, R_j)$ (See Figure 1 for examples). Then, we can define the metric $D$ between two representations as the Wasserstein distance (or optimal transport distance) (Villani, 2008; Peyré et al., 2019):

$$D(b_1, b_2) = \min_{\Gamma} \sum_{i,j} \Gamma_{ij} W_{ij}(b_1, b_2), \quad \text{s.t. } \Gamma \mathbf{1} = p(b_1), \ \Gamma^T \mathbf{1} = p(b_2) \tag{6}$$

For example, in Figure 1, the distance $D(b_1, b_2) = \frac{3}{2}, D(b_1, b_3) = \frac{3}{4}$ meaning $b_3$ is closer to $b_1$ than $b_2$. Hence the corresponding policy $\pi_3$ is closer to $\pi_1$ than $\pi_2$. Here, we provide a concrete method to measure

the distance between two solving processes. It is also provides a framework for general B&B algorithm. We can choose weight function $w$ and distance function $d$ depending on the property of the solution space and compute the distance between two B&B solving processes.

### 4.4 Novelty Search Evolutionary Strategy

Equipped with metric $D$ between representations, we can define the novelty score following Conti et al. (2018). Given a policy memory $M$ (a collection of older policies) and an instance $Q$ sampled from the problem distribution $\mathcal{D}$, novelty score is computed as:

$$N(\theta, Q, M) = \frac{1}{k} \sum_{\pi_j \in k\text{NN}(M, \theta)} D(b(\pi_\theta, Q), b(\pi_j, Q)) \tag{7}$$

where $k\text{NN}(M, \theta)$ is the $k$ nearest neighbor of $\pi_\theta$ in $M$. Back to Algorithm 1, B&B algorithm recursively splits the feasible region and obtains a set of polytopes when finishing solving an instance. Notice that a polytope in the set representation is invariant with the generating order, i.e. branching $x_1$ then $x_2$ will give the same polytope with branching $x_2$ then $x_1$. As a result, our metric $D$ and novelty score $N$ is mostly determined by the pruning behavior during the solving process. Put everything together, we summarize the training algorithm in section A.3.

## 5 Experiments

We now present comparative experiments against two competing machine learning approaches and three SCIP's branching rules to assess the value of our RL agent, as well as an ablation study to validate our choice of policy representation and training algorithm.

### 5.1 Setup

**Benchmarks**: We consider three classes of instances, Set Covering (Balas & Ho, 1980), Maximum Independent Set (Albert & Barabási, 2002) and Capacitated facility location (Cornuéjols et al., 1991), those are not only challenging for state-of-the-art solvers, but also representative for problems encountered in practice. For each class, we set up a backbone based on which we randomly generate the dataset as many real-world problems also share the same backbone. For example, a logistics company frequently solves instances on very similar transportation networks with different customer demands. We generate set covering instances using 1000 columns. We train on instances with 500 rows and evaluate on instances with 500 rows (test), 1000 rows (medium transfer), 1500 rows (hard transfer). We train maximum independent set on graphs with 400 nodes and evaluate on graphs with 400 nodes (test), 1000 nodes (medium transfer), and 1500 nodes (hard transfer). We generate capacitated facility location with 100 facilities. We train on instances with 40 customers (test) and evaluate on instances with 40 customers (test), 200 customers (medium transfer), and 400 customers (hard transfer). More details are provided in the section A.4

**Settings**: Throughout all experiments, we use SCIP 7.0.1 as the backend solver, with a time limit of 1 hour. For SCIP parameters, we have two settings: *clean* and *default*. The *clean* setting switches off other SCIP components, such as estimate node selection, cutting plane and primal heuristics. This way, the evaluation eliminates the interference from other components of the solver to variable selection policy. Under the *clean* setting, the solving nodes reflect the decision quality of variable selection policies only. So, we compare the decision quality of different methods under the *clean* setting. The *default* setting of SCIP will turn on all components inside SCIP, which is tuned for solving real problems. So, We compare the ability to solve challenging problems of different methods under the *default* setting.

**Baselines**: We compare against: Reliability Pseudocost Branch (RPB) (Achterberg & Berthold, 2009), the human-designed state-of-the-art branching rule, which computes strong branching in the beginning and

Table 1: Policy evaluation on test instances. Wins are counted by the number of times a method results in least number of solving nodes. The time $T_{\text{avg}}$ is reported in seconds.

| Method | $T_{\text{avg}}$ | $N_{\text{avg}}$ | Wins | $T_{\text{avg}}$ | $N_{\text{avg}}$ | Wins | $T_{\text{avg}}$ | $N_{\text{avg}}$ | Wins |
|--------|------|------|------|------|------|------|------|------|------|
| FSB | 99.73 | 367 | na/100 | 19.19 | 140 | na/100 | 27.16 | 964 | na/100 |
| RPB | 12.64 | 763 | na/100 | 3.06 | 250 | na/100 | 21.39 | 1449 | na/100 |
| VFS | 1935.35 | 737 | 5/ 75 | 244.14 | 1304 | 0/100 | 173.50 | 1848 | 31/100 |
| SVM | 21.19 | 856 | 1/100 | 10.83 | 498 | 1/100 | 29.64 | 2096 | 17/100 |
| GCN | 10.37 | 575 | 28/100 | 1.56 | 418 | 2/100 | 26.31 | 1752 | 13/100 |
| RL | **7.91** | **399** | **66**/100 | **1.26** | **200** | **97**/100 | **20.85** | **1640** | **39**/100 |
| | Set Covering | | | Independent Set | | | Facility Location | | |

gradually switches to simpler heuristics; Full Strong Branching (FSB), a full version of strong branching; Vanilla Full Strong Branching (VFS), strong branching with branch LP information muted (Gamrath et al., 2020); and two recent machine learning policies support vector machine (SVM) rank approach (Khalil et al., 2016) and GCN approach (Gasse et al., 2019) [1]. We denote our method as RL, which is the primal-dual net trained by NS-ES.

**Metrics**. To minimize the expected solving cost, metrics are selected as the average solving times ($T_{\text{avg}}$) for all instances and average solving nodes ($N_{\text{avg}}$) for instances solved by all methods. Since MIP instances could vary a lot in difficulty, we count the number of times each method leads the performance over the number of times each method solves the instance within timelimit (Wins) as a third robust metric.

**Implementation**. The detail of implementation is provided in section A.2

## 5.2 DECISION QUALITY

We evaluate the variable selection quality by solving 100 test instances under *clean* setting. Since we are comparing the decision quality, we say a method wins in this experiment if it results in the least number of solving nodes. As FSB and RPB benefit a lot from branching LP information (section 3), we do not include them when counting Wins. Table. 1 shows our RL agent leads the win times on all datasets and the average solving nodes on set covering, and independent set are significantly better than other methods.

## 5.3 GENERALIZATION TO LARGER INSTANCES

It is very important for RL agents to transfer to larger unseen instances as training on large instances is very expensive in the real world. We investigate the generalization ability of our RL agent by solving 100 transfer instances under *default* setting. To meet the needs in practice, we say a method wins in this experiment if it results in the fastest solving time. As VFS is not able to solve any transfer instance in time limit, we do not list its results in Table. 4. We can see, except for RPB and SVM having comparable performance on hard set covering and hard facility location, respectively, the RL agent leads the performance. In set covering (hard) and maximum independent set (hard), we do not compute the average number of nodes for full strong branching as it solves too limited instances.

## 5.4 IMPROVEMENT ANALYSIS

Having seen the improvements brought by RL, we would like to ask what kind of decisions our agent learns. We answer this question in two aspects: finding lower primal bound $c^*$ and obtaining higher dual value $\hat{c}$ that

---

[1]The source code has been released in Gasse et al. (2019).

Table 2: Policy evaluation on transfer instances. Wins are counted by the number of times a method results in fastest solving time. The time $T_{\mathrm{avg}}$ is reported in seconds.

| Method | $T_{\mathrm{avg}}$ | $N_{\mathrm{avg}}$ | Wins | $T_{\mathrm{avg}}$ | $N_{\mathrm{avg}}$ | Wins | $T_{\mathrm{avg}}$ | $N_{\mathrm{avg}}$ | Wins |
|---|---|---|---|---|---|---|---|---|---|
| | | | | | Medium | | | | |
| FSB | 1806 | 835 | 0/85 | 2835 | 95 | 0/57 | 506 | 208 | 3/100 |
| RPB | 360 | 10422 | 6/100 | 228 | **1354** | 8/100 | 311 | **562** | 4/100 |
| SVM | 1003 | 10170 | 0/92 | 1353 | 2687 | 8/81 | 314 | 1079 | 17/100 |
| GCN | 351 | 8789 | 7/100 | 1654 | 9292 | 0/76 | 332 | 1231 | 7/100 |
| RL | **295** | **7687** | **87**/100 | **148** | 1374 | **84**/100 | **258** | 1032 | **69**/100 |
| | | | | | Hard | | | | |
| FSB | 3361 | na | 0/15 | 3566 | na | 0/3 | 1046 | 101 | 0/90 |
| RPB | **1608** | 15043 | 30/80 | 1743 | **1618** | 21/82 | 763 | **281** | 5/99 |
| SVM | 2928 | 14058 | 0/30 | 3243 | 3850 | 0/20 | **606** | 623 | 37/100 |
| GCN | 1979 | 15043 | 0/71 | 2973 | 2114 | 0/26 | 891 | 865 | 5/97 |
| RL | 1628 | **12555** | **50**/80 | **1497** | 1648 | **62**/79 | 624 | 650 | **53**/100 |
| | Set Covering | | | Independent Set | | | Facility Location | | |

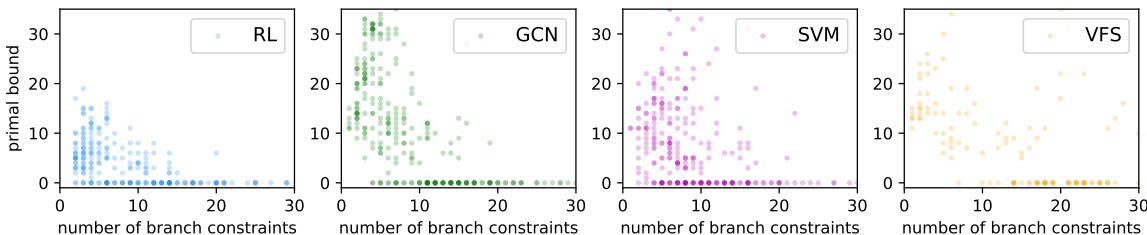

Figure 2: Primal bounds versus the depth in search tree (number of branch constraints) they are found

allows pruning in line 5 Algorithm 1. We compare our RL agent with GCN, SVM, VFC on 100 maximum independent set test instances under *clean* setting.

We first examine primal bound $c^*$. Figure 2 plots the feasible solutions found during the solving process. A point $(n, y)$ means we find a feasible solution $c^* = y$ in a subproblem containing $n$ branch constraints. Figure 2 shows that our RL agent is able to detect small $c^*$ at the early stage. Hence, it can prune more subproblems and solve the MIP faster. On the contrary, VFS fails to detect feasible solutions efficiently. One reason is, traditionally, strong branching or other human-designed heuristics are mainly on the purpose of obtaining higher $\hat{c}$. Our result suggests a new possibility for researchers to find variable selection method good at detecting feasible solutions.

Then, we check local dual value $\hat{c}$. To eliminate the influence in primal bound $c^*$ changing, we initialize $c^* = c_{\mathrm{opt}}$ with the optimal value like Khalil et al. (2016). We plot the curve of average width versus the depth in Figure 3. The area under the curve equals the average number of solving nodes, and we report it in the legend. Also, as $c^*$ is fixed, the width versus depth plot characterizes how many branches are needed to increase the local dual value $\hat{c}$ to $c^*$ so as to close a subproblem. A smaller width indicates the variable selection policy closes the gap faster. VFS performs better under this setting than in Figure 2 while it is still beat by learning based methods. Figure 3 shows that although our RL agent has the worst width in the beginning, it has the lowest peak and leads the overall performance. This means our RL agent successfully employs a non-myopic policy to maximize $\hat{c}$ in the long term.

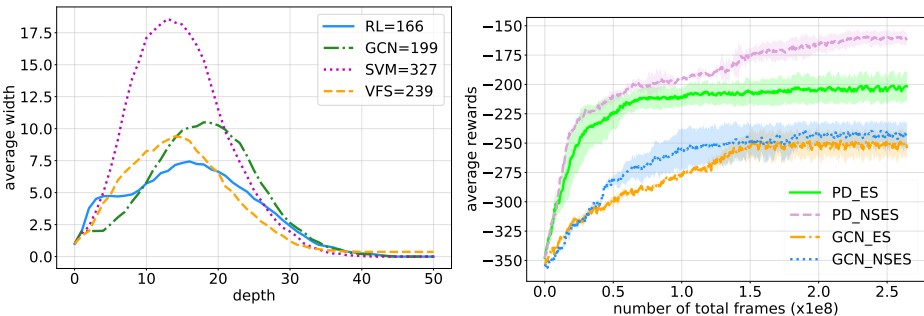

Figure 3: Average width versus depth          Figure 4: Comparison of 4 RL agents

## 5.5 ABLATION STUDY

We present an ablation study of our method on maximum independent set problem by comparing four types of RL agents: (1) PD policy + ES; (2) PD policy + NS-ES; (3) GCN + ES; (4) GCN + NS-ES. We sample $V = 200$ instances as our validation set in and plot the average number of solving nodes under clean setting on the validation set during the training process for five random seeds. All agents are initialized by imitation learning. The results are plotted in Figure 4. All curves obtain higher rewards shows that RL improves the variable selection policy. (1) and (2) having larger rewards than (3) and (4) shows that PD policy can obtain more improvement than GCN. Also, (2) and (4) having larger rewards than (1) and (3) shows that novelty search helps to find better policies. The results suggest that RL improves learning to branch and both PD policy, NS-ES are indispensable in the success of RL agent.

## 6 DISCUSSION

In this work, we point out the overestimation of the decision quality of strong branching. The evidence in Table 1 shows VFS performs poor on synthetic dataset under *clean* setting. An interesting phenomenon is that GCN can easily beat VFS after imitation learning (or our PD policy can obtain similar result). One possible explanation is that the primal-dual message passing structure naturally learns the good decisions and ignores the noise brought by strong branching. Another possible reason is the biased sampling. To keep the diversity of the samples, Gasse et al. (2019) employs a mixed policy of RPB and VFS to sample the training data. VFS probably performs good on most of the states while has poor decision quality when trapped in some regions. As a result, VFS has poor overall performance. Fortunately, using the mixing policy as the behavior policy helps to escape from these regions hence, the collected data have good decision quality. More studies are needed before we can give a confident answer for this question.

## 7 CONCLUSION

We present an NS-ES framework to automatically learn the variable selection policy for MIP. Central to our approach is the primal-dual policy network and the set representation of the B&B process. We demonstrate our RL agent makes high-quality variable selection across different problems types and sizes. Our results suggest that with carefully designed policy networks and learning algorithms, reinforcement learning has the potential to advance algorithms for solving MIP.

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

## A    APPENDIX

### A.1    BRANCH AND BOUND

Here we gives a simple illustration of B&B algorithm in Figure 5. Given the LP relaxation, the polytope represents the feasible region of the LP relaxation and the red arrow represents the objective vector. We first solve the LP relaxation and obtain the solution $\hat{x}$ as the red point. Noticing it is not feasible for MIP, we branch the LP relaxation into two subproblems. In (a) we select to split variable $x_1$ and in (b) we select to split variable $x_2$. The subproblems obtained after branching are displayed by the shaded purple regions. After finishing solve these two MIPs, we obtain the search trees $t_1$ and $t_2$. We can see that a wise selection of variable $x_2$ can solve the problem faster.

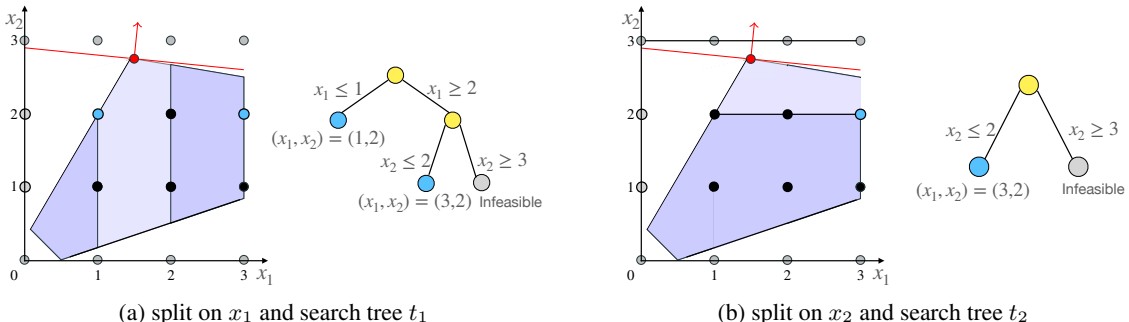

(a) split on $x_1$ and search tree $t_1$          (b) split on $x_2$ and search tree $t_2$

Figure 5: Illustration of splitting in B&B and the corresponding search tree

### A.2    IMPLEMENTATION

#### A.2.1    HARDWARE

All the experiments were run at a Ubuntu 18.04 machine with Intel(R) Xeon(R) Silver 4116 CPU @ 2.10GHz, 256 GB Memory and Nvidia RTX 2080Ti graphic cards.

#### A.2.2    PD POLICY

**Comparison**. PD policy is similar to the GCN in Gasse et al. (2019) but has two major differences. First, we use a dynamic reduced graph where fixed variables and trivial constraints are removed due to the variable bounds changing during the solving process while Gasse et al. (2019) do not consider it. The reduced graph can not only save computation, but also give a more accurate description of the solving state by ignoring the redundant information. The ablation in Section 5.5 shows it is indispensable in the success of RL. Second, we use a simple matrix multiplication in our PD policy while Gasse et al. (2019) use a complicated edge embedding in GCN. In some sense, GCN can be seen as an overparameterized version of our method. And our success reveals that message passing on the LP relaxation is the true helpful structure.

**detail**. We implement our primal dual policy net using dgl (Wang et al., 2019), with hidden dimension $h = 64$ and ReLU activation. The feature $\mathbf{X}$ for variable is a 17 dimension vector and feature $\mathbf{Y}$ for constraint is a 5 dimension vector. We list the detail of feature in Table. 3

| Tensor | Name | Description |
|---|---|---|
| **X** | type | a one-hot encoding for (binary, integer, implicit, continuous) |
| | coef | objective coefficient |
| | lb | variable lower bound |
| | ub | variable upper bound |
| | at-lb | indicator whether solution value equals lower bound |
| | at-ub | indicator whether solution value equals upper bound |
| | sol-frac | solution value fractionality |
| | basis-status | a one-hot encoding for simplex basis status (lower, basic, upper, zero) |
| | red | reduced cost |
| | age | normalized LP age |
| | sol-val | solution value |
| **Y** | obj-sim | cosine similarity with objective |
| | bias | bias value |
| | is-tight | tightness indicator in LP solution |
| | dualsol-val | dual solution value |
| | age | normalized LP age |

Table 3: Feature **X** for variable and feature **Y** for constraint

### A.2.3 BASELINE

**FSB**. We use the implementation in SCIP Gamrath et al. (2020)

**VFS**. We use the implementation in SCIP Gamrath et al. (2020)

**RPB**. We use the implementation in SCIP Gamrath et al. (2020)

**GCN**. We tried to implement GCN in dgl (Wang et al., 2019), however, it is significantly slower than the original implementation in Gasse et al. (2019). Hence, we still use the implementation in Gasse et al. (2019).

**SVM**. We use the implementation in Gasse et al. (2019).

### A.3 TRAINING

We have two settings *clean*, *default*. In experiments, we always train and test under the same setting.

**Imitation Learning**. We initialize our PD policy using imitation learning similar to Gasse et al. (2019). The difference is we only use 10000 training samples, 2000 validation samples and 10 training epochs as a warm start. In our setting, a policy from scratch can hardly solve an instance in a reasonable time, hence a warm start is necessary.

**Novelty Search Evolution Strategy**. We improve our RL agent using Algorithm 2. The parameters are set as $\alpha = 1e - 4$, $\sigma = 1e - 2$, $n = 40$, $V = 200$, $w = 0.25$, $\beta = 0.99$, $T = 1000$, $k = 10$.

---

**Algorithm 2:** Evolutionary Strategy with Novelty Score.

---

**Input:** Learning rate $\alpha$, Noise std $\sigma$, number of workers $n$, Validation size $V$, Batch size $M$, Initial weight $\lambda$, Weight decay rate $\beta$, Iterations T, Parameter $\theta_0$, Policy memory $M$, Instance distribution $\mathcal{D}$, Neighborhood size $k$.

**Output:** Best parameter $\theta_{\text{best}}$

1  Sample validation instances $Q_1, \cdots, Q_V \sim \mathcal{D}$
2  Set $R_{\text{best}} = \frac{1}{V} \sum_{j=1}^{V} R(\theta_0, Q_j), \theta_{\text{best}} = \theta_0$
3  **for** *t=0* **to** *T* **do**
4      Sample instances $P_1, \cdots, P_M \sim \mathcal{D}$
5      Sample $\epsilon_1, \cdots, \epsilon_n \sim \mathcal{N}(0, I)$ and compute $\theta_t^i = \theta_t + \sigma \epsilon_i$
6      Set $M = \{\theta_t^1, \cdots, \theta_t^n\}$
7      **for** *i=1* **to** *n* **do**
8          Compute $R_i = \frac{1}{m} \sum_{m=1}^{M} R(\theta_t^i, P_m)$
9          Compute $N_i = \frac{1}{m} \sum_{m=1}^{M} N(\theta_t^i, P_m, M)$
10     **end**
11     Set $\theta_{t+1} = \theta_t + \alpha \frac{1}{n\sigma} \sum_{i=1}^{n} \lambda \cdot N_i \epsilon_i + (1 - \lambda) \cdot R_i \epsilon_i$
12     Compute $R^{(t+1)} = \frac{1}{V} \sum_{j=1}^{V} R(\theta_{t+1}, Q_j)$
13     **if** $R^{(t+1)} > R_{best}$ **then**
14         Set $R_{\text{best}} = R^{(t+1)}, \theta_{\text{best}} = \theta_{t+1}, \lambda = \beta * \lambda$
15     **end**
16 **end**

---

### A.4   DATA SET

**Set Covering**. We generate a weighted set covering problem following Balas & Ho (1980). The problem is formulated as the following ILP.

$$\min \sum_{S \in \mathcal{S}} w_S x_S$$
$$\text{subject to} \sum_{S: e \in S} X_S \geq 1, \ \forall e \in \mathcal{U}$$
$$x_S \in \{0, 1\}, \ \forall S \in \mathcal{S}$$

where $\mathcal{U}$ is the universe of elements, $\mathcal{S}$ is the universe of the sets, $w$ is a weight vector. For any $e \in \mathcal{U}$ and $S \in \mathcal{S}$, $e \in S$ with probability 0.05. And we guarantee that for any $e$, it is contained by at least two sets in $\mathcal{S}$. Each $w_S$ is uniformly sampled from integer from 1 to 100.

We first generate a set covering problem with $\mathcal{U}_0 = \{e_1, \cdots, e_{400}\}$ and $\mathcal{S}_0 = \{S_1, \cdots, S_{1000}\}$ and set it as our backbone. Then, every time we want to generate a new problem with $m$ elements, we let $\mathcal{U} = \mathcal{U}_0 \cup \{e_{401}, e_{402}, \cdots, e_m\}$ add new $e_i$ into $S \in \mathcal{S}$ following the pipeline mentioned above.

**Maximum Independent Set**. We generate maximum independent set problem using Barabasi-Albert (Albert & Barabási, 2002) graphs. The problem is formulated as the following ILP.

$$\max \sum_{v \in V} x_v$$
$$\text{subject to} \quad x_u + x_v \leq 1, \ \forall e_{uv} \in E$$
$$x_v \in \{0, 1\}, \ \forall v \in V$$

where $V$ is the set of vertices and $E$ is the set of edges. We generate the BA graph using a preferential attachment with affinity coefficient 4.

We first generate a BA graph $G_0$ with 350 nodes. Then, every time we want to generate a new problem with $n$ variables, we expand $G_0$ using preferential attachment.

**Capacitated Facility Location**. We generate the capacitated facility location problem following Cornuéjols et al. (1991). The problem with $m$ customers and $n$ facilities is formulated as the following MIP.

$$\min \sum_{i=1}^{n} \sum_{j=1}^{m} c_{ij} d_j y_{ij} + \sum_{i=1}^{n} f_i x_i$$
$$\text{subject to} \sum_{i=1}^{n} y_{ij} = 1, \ \forall j = 1, \cdots, m$$
$$\sum_{j=1}^{m} d_j y_{ij} \leq u_i x_i, \ \forall i = 1 \cdots, n$$
$$y_{ij} \geq 0, \ \forall i = 1, \cdots, n \text{ and } j = 1, \cdots, m$$
$$x_i \in \{0, 1\}, \ \forall i = 1, \cdots, n$$

where $x_i = 1$ indicates facility $i$ is open, and $x_i = 0$ otherwise; $f_i$ is the fixed cost if facility $i$ is open; $d_j$ is the demand for customer $j$; $c_{ij}$ is the transportation cost between facility $j$ and customer $i$; $y_{ij}$ is the fraction of the demand of customer $j$ filled by facility $i$. Following Cornuéjols et al. (1991), where we first sample the location of facility and customers on a 2 dimension map. Then $c_i j$ is determined by the Euclidean distance between facility $i$ and customer $j$ and other parameters are sampled from the distribtuion given in Cornuéjols et al. (1991).

We first generate the location of 100 facilities and 40 customers as our backbone. Then, every time we want to generate a new problem with $m$ customers, we generate new $m - 40$ locations for customers and follow the pipeline mentioned above.

## A.5 EXPERIMENTS ON BENCHMARK FROM GASSE ET AL

We are mostly interested in improving the variable selection policy on similar problems hence, we generate our benchmark based on a backbone. The backbone allows the instances share some common structures such that there exists a good policy for the given distribution of problems. Our experiments show that NS-ES is able to learn a good policies on this purpose. However, it is also interesting to check the performance of our method on a more random distribution. Here, we conduct experiments on benchmark from Gasse et al. (2019). We employ the same instance generator and SCIP setting as Gasse et al. (2019). For each category, we evaluate the policy on 20 instances with 5 random seeds. We report the average solving time $T_{\text{avg}}$ and shifted geometric mean solving time $T_{\text{geo}}$ on all instances, average solving nodes $N_{\text{avg}}$ and shifted geometric mean solving nodes $N_{\text{geo}}$ on instances solved by all instances, and Wins for the times one methods leads

Table 4: Policy evaluation on benchmark from Gasse et al. (2019)

| Method | Easy | | | | | Medium | | | | | Hard | | | | |
|---|---|---|---|---|---|---|---|---|---|---|---|---|---|---|---|
| | $T_{avg}$ | $T_{geo}$ | $N_{avg}$ | $N_{geo}$ | Wins | $T_{avg}$ | $T_{geo}$ | $N_{avg}$ | $N_{geo}$ | Wins | $T_{avg}$ | $T_{geo}$ | $N_{avg}$ | $N_{geo}$ | Wins |
| VFS | 155.74 | 74.84 | 249 | 122 | 0/100 | | | | | | | | | | |
| RPB | 10.94 | 9.57 | 284 | **58** | 0/100 | 121.3 | 76.8 | 7351 | 2413 | 0/100 | 2406 | 2038 | 50702 | 46487 | 9/56 |
| SVM | 14.16 | 11.49 | 348 | 166 | 0/100 | 220.8 | 108.1 | 6786 | 2522 | 0/100 | 3140 | 3029 | 55052 | 52059 | 0/34 |
| GCN | 8.84 | 7.97 | 282 | 132 | 5/100 | **88.5** | 53.1 | **4614** | **1844** | 44/100 | **2311** | **1971** | 39735 | 37699 | **52**/70 |
| RL | **7.54** | **6.38** | 275 | 133 | **95**/100 | 96.4 | **51.5** | 4761 | 1872 | **56**/100 | 2611 | 2248 | **39802** | **36863** | 9/51 |

Set Covering

| Method | $T_{avg}$ | $T_{geo}$ | $N_{avg}$ | $N_{geo}$ | Wins | $T_{avg}$ | $T_{geo}$ | $N_{avg}$ | $N_{geo}$ | Wins | $T_{avg}$ | $T_{geo}$ | $N_{avg}$ | $N_{geo}$ | Wins |
|---|---|---|---|---|---|---|---|---|---|---|---|---|---|---|---|
| VFS | 103.32 | 81.77 | 121 | 84 | 0/100 | | | | | | | | | | |
| RPB | 3.39 | 3.10 | **29** | **10** | 0/100 | 20.2 | 19.3 | 1036 | 732 | 0/100 | 243 | 167 | 43147 | 9074 | 5/100 |
| SVM | 3.09 | 2.71 | 125 | 78 | 0/100 | 30.9 | 26.3 | 1189 | 867 | 0/100 | 420 | 272 | 17171 | 10933 | 0/100 |
| GCN | 2.26 | 2.15 | 106 | 69 | 29/100 | 15.0 | 13.1 | 964 | 694 | 46/100 | 219 | 138 | 13046 | 7661 | 29/100 |
| RL | **2.16** | **2.00** | 107 | 68 | **71**/100 | **14.3** | **12.8** | **876** | **665** | 54/100 | **212** | **132** | **12700** | **7531** | **66**/100 |

Conbinatorial Auction

| Method | $T_{avg}$ | $T_{geo}$ | $N_{avg}$ | $N_{geo}$ | Wins | $T_{avg}$ | $T_{geo}$ | $N_{avg}$ | $N_{geo}$ | Wins | $T_{avg}$ | $T_{geo}$ | $N_{avg}$ | $N_{geo}$ | Wins |
|---|---|---|---|---|---|---|---|---|---|---|---|---|---|---|---|
| VFS | 298.52 | 94.18 | 399 | 122 | 0/100 | | | | | | | | | | |
| RPB | 50.84 | 29.18 | **341** | **25** | 13/100 | 219.9 | 165.5 | **371** | **151** | 13/100 | 863 | 624 | **245** | **109** | 19/97 |
| SVM | 84.06 | 33.71 | 571 | 115 | 3/100 | 209.3 | 146.9 | 606 | 343 | 16/100 | 1365 | 1029 | 513 | 334 | 1/91 |
| GCN | 71.60 | 27.39 | 684 | 109 | 21/100 | 196.8 | **133.0** | 586 | 344 | 28/100 | 876 | **573** | 483 | 309 | **48**/95 |
| RL | **56.43** | **25.00** | 538 | 109 | **63**/100 | **191.9** | 136.7 | 577 | 327 | **43**/100 | **807** | 583 | 527 | 367 | 29/95 |

Capacitated Facility Location

| Method | $T_{avg}$ | $T_{geo}$ | $N_{avg}$ | $N_{geo}$ | Wins | $T_{avg}$ | $T_{geo}$ | $N_{avg}$ | $N_{geo}$ | Wins | $T_{avg}$ | $T_{geo}$ | $N_{avg}$ | $N_{geo}$ | Wins |
|---|---|---|---|---|---|---|---|---|---|---|---|---|---|---|---|
| VFS | 385.68 | 144.61 | 205 | 50 | 0/100 | | | | | | | | | | |
| RPB | 11.68 | 10.24 | **270** | **21** | 15/100 | 185.3 | 143.8 | **5687** | 2437 | 7/100 | 2809 | 2365 | 8374 | 7294 | 4/38 |
| SVM | 20.67 | 10.78 | 431 | 39 | 1/100 | 723.5 | 306.8 | 8692 | 2595 | 2/100 | 3330 | 3159 | 12991 | 10807 | 0/14 |
| GCN | 10.92 | 8.96 | 333 | 37 | 0/100 | **161.6** | **87.4** | 11110 | 2563 | **50**/100 | **2422** | **1500** | **3236** | **2607** | 37/43 |
| RL | **10.38** | **7.64** | 305 | 38 | **84**/100 | 240.9 | 104.4 | 9767 | **2082** | 41/100 | 3039 | 2439 | 9241 | 5936 | 8/23 |

Maximum Independent Set

over the number of instance solving to optimal. As VFS is too slow to solve challenge instances, we only report its performance on easy instances.

We can see that, in Table 4, the improvement from RL method is less than that in the main text. Intuitively, the randomly generated instances have less shared structure and leave less room for RL to improve the policy. How can we improve branch policies for randomly generated problems is still a question needs more explorations in the future.

