# OpenReview forum: "Improving Learning to Branch via Reinforcement Learning"
_ICLR.cc/2021/Conference — Reject_

### Official Review · AnonReviewer1 · 2020-10-28
**Interesting ideas, but experiments appear unsound**

**Rating:** 4
**Confidence:** 5

**Review:**

###############################################################################

Summary

The paper presents a novel method for learning branching strategies within branch-and-bound solvers, which consists in a graph-convolutional network (GCN) combined with a novelty-search evolutionary strategy (NS-ES) for training, and a new representation of B&B trees for computing novelty scores. The paper claims this new method provides a significant improvement over state-of-the-art branching strategies, either based on expert-designed rules or on imitation learning of strong branching.

###############################################################################

Pros and cons

Pros:
 - the idea of learning branching strategies using reinforcement learning instead of imitation learning makes a lot of sense and seems like a promising direction to follow
 - the proposed representation of B&B trees is original
 - the presented results seem promising

Cons:
 - the experimental setup followed in the paper is questionable
 - the presented experimental results are inconsistent with the literature, which I find suspicious
 - the overall method description comprises several blind spots and fallacious arguments

###############################################################################

Recommendation

While I like the idea pursued in the paper, and I believe the proposed method might be promising, the paper is mostly experimental, and is not sound enough on that side. I therefore recommend rejection of the paper.

First, I found the reported experimental results suspicious. The paper is mostly based on the work from Gasse et al., reuses the same code, (almost) the same problem benchmarks, and some portions of text which are identical. This is fine, but then why did the authors conduct experiments only on 3 out of the 4 available benchmarks ? And why are the results inconsistent with those reported in the original paper ? In the presented experiment the SVM model consistently performs better than the GCN model in terms of number of nodes. Why is it so, if literally the same code has been reused, as mentioned in the appendix ? This alone raises serious doubt in my mind about the validity of the reported numbers. At the very least the authors must provide an explanation for that. Also, Figure 4 indicates that a pre-trained GCN model results in a tree size of 350 on independent set problems, which does not coincide with the number 418 reported in Table 1.

Second, the experimental setup itself is questionable. The authors consider two setups for the SCIP solver, "clean", with depth-first-search node selection and all other functions disabled (whatever that means), and "default". No explanation is given as to why those two setups are considered. Some experiments are conducted under the "clean" setup (Table 1, Figure 2 and 3), other are made under the "default" setup (Table 2), and for some experiments this information is missing (Figure 4). What is the purpose of the "clean" setup ? It is known that node selection and branching strategies do interact with each other, and all the evaluated branching methods were proposed in the context of a default, state-of-the-art solver. Why then comparing such methods in the "clean" setting ? What is the justification here ? Also, the paper lacks a proper ablation study. How much of the reported improvements come from the policy model ? From the RL training ? From the proposed novelty measure ?

Third, I found several arguments to be fallacious. The authors claim their method results in non-myopic policies, without defining what they mean by that. Assuming they mean the policies have access to non-local information, they later on contradict this claim, by acknowledging that the proposed model only processes the local node's LP-based information. The authors also argue that imitating strong branching (SB) is not a good idea, as it results in small trees not because of its branching decisions, but because of side-effects. To back-up their claim, the authors present numbers showing that, once those side effects are removed, the SB tree size is much higher. But in the same table, the GCN model trained to imitate SB results in trees much smaller than the expert (418 vs 1304 on independent set), which again is suspicious, and most importantly contradicts the original claim of the authors. Given those numbers, imitating SB looks like a good idea.

Finally, the proposed method comprises blind spots. The authors present their model as a primal-dual iteration policy, for what looks like a simplified GCN architecture for the one from Gasse et al. The same structure (bipartite graph) and the same features are used. The authors present a Lagrangian dual LP formulation in Equation 4, however I do not see the connection to the proposed model. How is the proposed model a primal-dual iteration policy ? The authors then present what is I believe the most original contribution of the paper, a distance metric for branch-and-bound trees. However, again I found the description of the method rather sloppy. Counting the number of integral points inside a polytope is a hard problem in itself. Is the counting performed over box relaxations only ? If so, how can the proposed representation process MILPs with unbounded variables ? This seems to be a strong limitation to me, which at least deserves a discussion.

###############################################################################

Questions to authors

I would appreciate if the authors could clarify why they conduct some of their experiments in the "clean" setting, why they don't report results on the complete benchmark from Gasse et al., and also comment on the performance of SVM vs GCN, which contradicts what is reported in Gasse et al.

###############################################################################

Final recommendation

I thank the authors for their detailed response. The authors have clarified some technical details and blind spots of their methods, have fixed their evaluation metric which was wrongfully comparing tree sizes on solved and unsolved instances, and have presented an additional experiment in the appendix on the original benchmark from Gasse et al. These changes are going in the right direction. However, I remain concerned about the experimental setup in the paper, and therefore my final recommendation is still rejection.

First, I still find suspicious that the main experiment in the paper is conducted on only 3 (modified) benchmarks out of the 4 proposed in Gasse et al. The authors claim they did not run experiments on the 4th benchmark due limited computational resources, but at the same time they present complete results on the 4 original benchmarks in the appendix. Therefore I believe the explanation given by the authors is fallacious. Results on the 4th (modified) benchmark, and even better, on other additional benchmarks, would be much more convincing and alleviate any doubt about cherry-picking.

Second, I am not convinced by the argument the authors present to justify their two solver settings: clean and default. I do agree that challenging problems should be solved under the default setting, however I do not see why decision quality should be measured using the clean setting. Decision quality matters in the default setting as well, and one could argue it matters in the default setting only, if the final goal is to improve the solving time of the solver on challenging problems. Moreover, if the authors want to use the tree size as a mean to measure branching decision quality, they must also provide the optimal solution value to the solver at the beginning of the solving process, in order to deactivate side-effects from pruning. See G. Gamrath and C. Schubert, 2018, "Measuring the Impact of Branching Rules for Mixed-Integer Programming".

Last, the method proposed by the authors seems to be effective only in the specific benchmark they propose. In the additional experiments they present in the appendix (Table 4), their method does not convincingly improve over the original method from Gasse et al., as the performance gains on the Easy training instances degrade rapidly as one moves away towards the more challenging Hard instances. The very name of the paper, "Improving Learning to Branch via Reinforcement Learning", seems to claim that reinforcement learning improves existing learning to branch methods. However, the improvement observed by the authors is very specific to the "backbone" setting they propose, and does not seem to translate to the original benchmarks of the methods they compare to. The authors justify their choice of benchmarks saying "we focus on a more realistic industrial setting", but I am unsure whether the hypothesis of a "backbone graph" is particularly realistic in industrial applications. I do not think the original benchmark from Gasse et al. is particularly realistic either, however I would not give more or less value to either one of the two. As such, I believe it is crucial that the authors report experimental results on both benchmarks in the main body, and provide a discussion as to why RL seems to bring improvement in the restricted "backbone graph" benchmarks, but not in the original ones which have more variability. This, in my opinion, would have a much higher scientific value than simply presenting both a new method and a new benchmark, while disregarding how the method performs on previous benchmarks from the literature.

In light of the changes made by the authors I am willing to raise my rating, however I still recommend rejection for ICLR.

###############################################################################

Additional feedback

p.2 §2: a linear programming -> a linear program

p.2 §2: efficiency-effectiveness trade-off -> What do you mean by efficient and effective ? What is the difference ? Do you mean the decision quality / computation time trade-off ?

p.2 §3: ignores the redundant information and makes high-quality decisions on the fly -> This is quite vague. How does your policy achieve that ? What redundant information are you talking about ? What is the efficiency-effectiveness trade-off ?

p.2 §3: For exploration, we introduce a new representation of the B&B solving process -> How is this new representation related to exploration ? I am missing the argument here.

p.2 §5: set covering, maximum independent set, capacitated facility location -> In Gasse et al. 2019, to which you compare as a baseline, there is a fourth benchmark, combinatorial auctions. I must say that not reporting experiments on this complete benchmark is suspicious, as I do not see a reason for not doing it.

p.2 §4: primal-dual policy -> Where does this name come from ? Why is your policy primal-dual ?

p.2 §5: the long-term overestimation of strong branching -> What does that mean ?

p.2 §5: based on primal-dual iteration over reduced LP relaxation -> What does that mean ?

p.3 §2: in line 2 -> line 3 I believe

p.3 §2: visulization -> visualization

p.3 Equation 3: I am not sure I understand this formula. I suppose that $\phi_t=(\theta_t,\sigma_t)$ ? Then $\sigma_t$ is never updated, but only $\theta_t$ is ? What is $\epsilon_i$ here ? I suggest that you clarify the meaning of this equation, which I could understand only after reading Wierstra et al. 2014, Natural Evolution Strategies. Making it explicit what it corresponds to (expected gradient $\frac{\delta R(\theta)}{\delta \theta}$ over the population) would greatly help the reader. At the very minimum, all terms should be defined properly.

p.3 §4: if strong branching finds [...] -> I hardly understand that sentence. Do you mean that strong branching will induce additional domain propagation (Achterberg 2007, Constraint Integer Programming, §2.3) ?

p.4 §2: we set the reward $r_t = -1$ with discount $\gamma = 1$ -> Maximizing such a reward is equivalent to minimizing the B&B tree size. And as a result it aligns with your evaluation metric (nodes). This should be mentioned, again, for clarity.

p.4 Section 4.2: Primal Dual POlicy Net -> I don't really understand this primal-dual thing, given that you use the same features as in Gasse et al. (Table 3) and just replace their GCN model by a simpler, underparameterized version (A.2.2).

p.4 Equation 4: I do not see the point of introducing a Lagrangian relaxation here. Neither Equation (4) nor $\lambda$ are ever used or referred to in the text.

p.4 §4: $f_\mathcal{C}, f_\mathcal{C}$ -> $f_\mathcal{C}, f_\mathcal{V}$

p.4 §4: one layer -> one hidden layer ?

p.4 §4: For efficiency, we do not include problem set $S$, which makes it a partial observable MDP -> Then your model only has access to local information, and as such results in myopic policies, which contradict one of your initial claims. Myopic by definition means your vision is limited, which is the case here if your model can not observe the full state of the solver but only the local LP.  Unless you mean something else with "myopic", which does not have a formal definition in the paper. I think you confuse myopic policies with greedy policies.

p.5 §1: as the collection of those leaf subproblems -> Which leaf subproblems ? The leaves of the complete B&B tree ? The partial tree ?

p.5 §2: $w(R_i):= \dots$ -> I understand $Q$ is the original MILP, while $R_i$ is a box. $Q$ restricted to $R_i$ is therefore a regular MILP as well (the local MILP). Counting the number of feasible solutions for a MILP is I believe an NP-hard problem. How do the authors afford to do that ? Do you mean that you look for integral solutions within $R_i$ ? A second comment here: What if some variables in Q are unbounded ? Your weight function does not seem to handle that case...

p.5 §2: the feasible solution -> a feasible solution ?

p.5 §3: Notice that a polytope in the set representation is invariant with the generating order [...] -> This sentence is ambiguous, and is simply not true. Branching on $x_1$ then $x_2$, depending on whether it is the left child or the right child which is subsequently branched on, does not yield the same collection of polytopes.

p.5 §3: pruning behavior -> What is meant here by pruning behavior ? Node selection ? A common procedure to compare branching strategies, is to removes potential side-effects from node selection by providing the algorithm with the optimal objective value from the start. See, e.g., Gamrath and Schubert, 2017, Measuring the impact of branching rules for mixed-integer programming. In this scenario where the goal of B&B is just to close the dual gap, the branching strategy remains a major component of B&B.  This seems to contradict your point here, that your novelty measure is both relevant for branching, while it is mostly driven by the pruning behavior.

p.6 §3: we have two settings -> Why having those two settings ? What is the point of the "clean" setting ? Also, I think you mean depth-first-search, not deep-first-search.

p.6 §7: under clean setting -> Why under that setting ? Would the "default" setting be more representative for evaluating the performance of branching strategies ? It is known that branching interacts a lot with node selection, which you arbitrarily changed here to depth-first-search. Is there a reason for that ?

p.7 Table 1: I suggest that you group FSB and RPB together, since they can not be compared to the other methods in terms of number of nodes (unfair node counting). Also, the RPB $N_avg$ value should not be bolted, since RPB can not be compared to here in terms of nodes.

p.7 §1: under the default setting -> Why do you suddenly switch to the "default" setting ?

p.7 Table 2: Those numbers are very doubtful. Why does the GCN model result here in larger trees than the SVM model, while the opposite is observed in Gasse et al. ? This should, at the very least, be discussed.

p.7 §3: founded -> found

p.8 §2: Then, we check dual value $\hat{c}$ -> If you want to assess the capacity of each strategy for closing the dual gap, why not simply reporting the evolution of $\hat{c}$ over time ?

p.8 §2: our RL agent successfully employs a non-myopic policy to maximize $\hat{c}$ in the long term -> I do not see that from the curves... I only see that the tree size is smaller, and distributed differently that with the other methods.

p.8 Section 5.5: This ablation study is missing a key ingredient: what is the performance of the PD model, trained via imitation learning ? Which part of your improvements comes from your PD model ? Which part comes from RL ?

p.8 Figure 4: Why don't those numbers align with those in Table 1 ? Is the ablation study conducted in the "clean" setting ? Why do all curves in Figure 4 start at 350 nodes, when the optimal GCN model in Table 1 is at 418 nodes ?

---

> ### Author Response · Authors · 2020-11-18
> **Response 1**
>
> Dear reviewer, thanks for reading our work and giving your valuable suggestions.
>
> why did the authors conduct experiments only on 3 out of the 4 available benchmarks?
>     (1)We have limited computational resource and we thought one covering problem, one packing problem, and one mixed integer programming problem are representative.
>     (2)We update the experiments on benchmark from Gasse et al. in the appendix.
>
> Figure 4 indicates that a pre-trained GCN model results in a tree size of 350 on independent set problems, which does not coincide with the number 418 reported in Table 1.
>     In Table 1, the evaluation is performed on the test set. In Figure 4, the evaluation is performed on a valid set. That's why they have different numbers.
>
> 'clean' and 'default' setting
>     (1)The 'clean' setting switches off other SCIP components, such as estimate node selection and cutting plane. This way, the evaluation eliminates the interference from other components of the solver to variable selection policy. Under the 'clean' setting, the solving nodes reflect the decision quality of variable selection policies only. So, we compare the decision quality of different methods under the 'clean' setting.
>     (2)The 'default' setting of SCIP will turn on all components inside SCIP, such as estimate node selection and cutting plane, which is tuned for solving real problems. So, We compare the ability to solve challenging problems of different methods under the 'default' setting.
>
>
> ablation
>     (1) In Table 1, we report the performance on test data while in Figure 4, we report the performance on validation data. That's why they give different numbers of nodes.
>     (2) We initialize the policy by imitation learning. Our method leads to higher rewards and shows the benefit of our RL training method.
>     (3) PD+ES is better than GCN+ES, PD+NES is better than GCN+NES. It shows the benefit of the policy model.
>     (4) PD+NES is better than PD+ES, GCN+NES is better than GCN+ES. It shows the benefit of the novelty score.
>
> myopic vs nonmyopic
>     (1) The 'myopic' or 'nonmyopic' here refers to the attention on immediate reward or the long term reward, respectively, rather than the observation of the whole state.
>     (2) 'myopic' means the policy pursues a large immediate reward. For example, strong branching maximizes the dual bound improvement in one variable selection.
>     (3) 'nonmyopic' means the policy pursue a large long term reward. For example, in our MDP formulation, our policy tries to minimize the size of the search tree.
>
> Is strong branching powerful?
>     (1) The result in table 1 shows that if we disable its side effects, such as domain reduction and conflict analysis, strong branching (SB) will result in a large search tree. This supports our claim.
>     (2) We also observe an interesting phenomenon that a policy can easily beat SB after imitating SB. We propose our guesses in appendix A.5. One possible reason is the policy net has good architecture such that it can easily learn good decisions. Another explanation is the off-policy sampling. For example, in Gasse's sampling phase, the behavior policy is a mixture of reliable pseudo cost branching and strong branching. The exact reason for this phenomenon is not clear yet and we believe figure this out can help improve designing or learning a variable selection policy.
>
> The primal-dual policy
>     (1) We call it primal-dual policy as our policy net is inspired by the primal-dual iteration for solving a constrained linear program.
>     (2) The first step is to reduce the state. Our state is a relaxed LP, which has a mapping to a primal-dual iteration on its Lagrangian relaxation. In the view of the optimization, fixed variables (lower bound equals to upper bound) and trivial constraints (always true) are redundant. They do not pass any information (always zero) in the primal-dual iteration. So, we call them redundant information. Removing redundant information, we obtain our reduced graph. This reduction not only simplifies the state representation but also saves the computation.
>     (3) The primal-dual iterations between variables and constraints can be seen as a message passing. We insert neural network layers in message passing and obtain our policy network.
>
> Counting as weight function in novelty score:
>     (1) Here, we propose a general framework for computing the distance between two partitions of a set. The weight function and distance function are not fixed. For different solution sets, we can choose other weight functions. For example, when the solution set is unbounded, the weight function could be chosen as a Gaussian measure instead of a uniform measure (counting).
>     (2) In this work, we select the weight function as counting in the box relaxation as it is simple and performs well in our experiments.

---

> ### Author Response · Authors · 2020-11-18
> **Response 2**
>
> p.2 §3 a new representation of the B&B solving process
>     (1) The representation here means the partition of the box relaxation of the feasible region.
>     (2) In the novelty search framework, we give credits to a policy far from the current population to encourage exploration. For variable selection policies in the B\&B algorithm, measuring the distance between the two policies is not straightforward. So, we measure the distance between two policies by measuring how they partition the solution set.
>
> p.2 §5 the long-term overestimation of strong branching ... What does that mean?
>     (1) In the previous method, people imitate strong branching as it always results in the smallest search tree. We point out that the reason strong branching produces a small search tree is the side effect in solving branch LP. This can not be learned by a variable selection policy. That's why we say the decision quality of strong branching is overestimated.
>
> p.3 §4 if strong branching finds [...]
>     (1) What we meant here is strong branching include additional conflict analysis (https://www.sciencedirect.com/science/article/pii/S1572528606000818).
>
>
> p.5 §3 A polytope in the set representation is invariant with the generating order
>     (1) What we meant here is different variable selection order probably generate the same partition. For example, given a box $[0,1]^3$, we show two generating order: (A) We first split $x_1$, then split $x_2$ on left child and right child. It results in a partition $\{\{0\} \times \{0\} \times [0,1], \{0\}\times\{1\}\times[0,1], \{1\}\times\{0\}\times[0,1], \{1\}\times\{1\}\times[0,1]\}$; (B) We first split $x_2$, then split $x_1$ on left child and right child. It results in a partition $\{\{0\} \times \{0\} \times [0,1], \{1\}\times\{0\}\times[0,1], \{0\}\times\{1\}\times[0,1], \{1\}\times\{1\}\times[0,1]\}$. These two generating orders give rise to the same partition.
>
>
> p.5 §3 pruning behavior
>     (1) Pruning behavior is also a result of variable selection. A better variable selection can increase the local dual bound faster, hence we can prune the subtree earlier. See figure.5 as an example. Splitting $x_2$ results in a smaller search tree than splitting $x_1$.
>
> p.8 §2 Then, we check dual value $\hat{c}$. If you want to assess the capacity of each strategy for closing the dual gap, why not simply reporting the $\hat{c}$ evolution overtime?
>      (1)In this experiment, we track the local dual values versus depth rather than the global dual bound over time. Since the optimal primal bound is given at the beginning and the duality gap will be zero eventually, we care more about how many branches are needed to close a subproblem rather than the dual value gain after one decision. In this sense, width versus depth is able to provide more information.
>     The width at depth $d$ characterizes how many subproblems remain open (local dual values lower than optimal value) after adding $d$ branch constraints. A greedy variable selection may close some subproblems fast while leaving others being very hard to solve.
>     We say our method is 'nonmyopic' as it allows most of the sub-problems to be solved in reasonable steps (low peak and light tail). We think this idea also applies in a strong branching score, where the product of left gain and right gain usually performs better than the summation of two gains.

---

### Official Review · AnonReviewer3 · 2020-10-28
**Solid contribution and interesting observations**

**Rating:** 7
**Confidence:** 3

**Review:**


### Summary

The paper proposes a model for *variable selection* in *Mixed Integer Programming (MIP)* solvers. While this problem is clearly a sequential decision making task, modeling it as an MDP is challenging. As a result, existing works use other approaches such as ranking or imitation learning. This paper overcomes these challenges by introducing a new problem representation.

Additionally, the paper makes an interesting observation: It demonstrate the problem with mimicking an existing heuristic. This approach has been used in recent works on learning to branch.

### Strong points

- Introducing the MDP formulation is a significant and timely contribution
- The writing and presentation is very good
- The observation about imitating strong branching can spark follow-up investigation
- The empirical results are promising

### Aspects to improve

- The benchmarks and results do not match those reported in *(Gasse et al., 2019)*. This makes it difficult to compare the results of the two papers. I suggest adding experiments using those instances in the appendix.

### Recommendation

This paper introduces a new direction in the area of *learning to branch in MIP solvers*.  The empirical evaluation suggests promise for the proposed method. The paper has potential for follow-up work. I recommend accepting the paper.

---

> ### Author Response · Authors · 2020-11-18
> **Response**
>
> Dear reviewer, thanks for reading our work and giving your valuable suggestions.
>
> We add experiments on benchmark from Gasse et al. in the appendix.

---

### Official Review · AnonReviewer2 · 2020-10-28
**Great work!**

**Rating:** 7
**Confidence:** 3

**Review:**

### Summary

The authors present a way of achieving high-quality variable-selection decisions inside a branch-and-bound algorithm for MIP Solving. The learning happens via evolutionary search with a bias for novelty which the authors introduce.

### Reasons for score

The authors show a significant application performance improvement compared to existing learning aproaches, and visualization/ablations showring where the improvements may be coming from. The paper is readable.


### Pros

1. Clear and readable
1. The experimentation is exhaustive: different MIP Solving settings (e.g. "clean and default") and unseen test set problem-instance sizes generalization in the size dimension.
1. Visualization to try to understand where the improvement is coming from.
1. Ablations of novelty component inside search and learning architecture

### Cons/Clarifications

1. I don't feel confident that the evidence presented that strong-branching decisions are bad, because of the non-diverse and small sized data distribution -- section 5.2 Table 1 -- I would feel more confident if the data were 300 randomly sampled instances from MIPLIB *if you don't get timeouts in your evaluation then it isn't large enough to be interesting application-wise)
1. the reward described may not work well when training on instances large enough that take lots of steps to solve
1. the PD policy net is a graph neural network specialized for bipartite graphs with single-scalar-attributed edges, and its receptive field is 1 hop per layer, so 2 hops for the architecture used in the paper, which raise the question: how does the diameter (or average variable-variable distance) of the data distribution compare to that?
1. I don't really know if the novelty score will scale well to large MIP instances, or if it is defined for infeasible MIPs. (The network could still be trained on small feasible MIPs).

### Other questions

1. Figure 2 shows that the primal bound improves with the RL network, can we know why/how this is happening? Having a better dual bound could lead to pruning based on that and thus having a better primal bound.

### Typos

founded -> found in page 7 section 5.4 https://pasttenses.com/find-past-tense

---

> ### Author Response · Authors · 2020-11-18
> **Response**
>
> Dear reviewer, thanks for reading our work and giving your valuable suggestions.
>
> (1) Yes, I agree that our experiments only show that strong branching (with side effects deactivated) has poor decision quality on our dataset. I test the instance air04, air05 on my machine in the default setting. Relpscost branching is able to solve them in 125, 658 nodes, full strong branching (with side effects activated) is able to solve them in 23, 51 nodes, vanilla full strong branching (with side effects deactivated) is able to solve them in 51, 163 nodes, and random branching is able to solve them in 15283, 11773 nodes. This shows (1) vanilla full strong branching has good decision quality on these two instances. (2) MIPLIP instances might have very different properties compared with our synthetic dataset. Based on this observation, we do need more experiments on MIPLIB to examine the decision quality of vanilla full strong branching on MIPLIB.
>
> (2) Yes, for large instances we need to clip the reward. For natural evolutionary strategy, rank-based fitness shaping is a commonly used trick to normalize the reward. Although not technically sound, the trick works well empirically.
>
> (3)  In our dataset, the diameter for set covering is 2, for capacitated facility location is 2, for the maximum independent set is 5.
>
> (4) Yes, it is true. For large instances, computing the novelty score requires intensive computation as we introduce optimal transport. Ideally, we hope we only need to train the policy on small instances, as rolling out a trajectory on the large instance itself is very hard.
>
> (5) Yes, the phenomenon in Figure 2 is interesting. RL policy provides a better balance between two children such that avoids obtaining large gain in one child without good primal solution while obtaining small gain in another child containing a good primal solution.

---

> > ### Comment · AnonReviewer2 · 2020-11-24
> > **Thank you for response**
> >
> > Thank you authors for the response.
> >
> > I am happy with the answers to each of my points.
> >
> > (3) It is possible on datasets with MIPs with an average diameter length that is (much) higher than the current ones may require graph neural networks with a higher number of message passing steps. This might mean that to generalize to several orders of magnitude in the size of the input graph diameter, the network needs to be repeated enough times so the receptive field is high enough.

---

### Official Review · AnonReviewer4 · 2020-10-28
**Interesting and successful approach to branch selection**

**Rating:** 8
**Confidence:** 3

**Review:**

The paper proposes a method of choosing variables for branching in branch and bound approaches to MIP solving. The approach is based on reinforcement learning.

The paper presents an adequate overview of previous approaches to the problem. There is not a lot of detail about how these approaches work but the overview of the techniques given allows the reader to see how the proposed approach differs from this earlier work and motivates the technique.

The MDP formulation used appears not to be novel, but the paper presents a novel way to use reinforcement learning to find good strategies.

The paper makes a clear argument about why simply choosing the same variable as strong branch would is not the best variable selection strategy. This is supported by later experimental results. I am not sufficiently close to the field to know whether this is a novel argument or accepted fact. In any case, it seems worth stating here as it illustrates a significant fault with many existing alternative approaches.

There are quite a lot of decisions made in the design of the algorithm. While it is clear what has been done, I am unclear on why many of these choices have been made. For example, what is it about NS-ES that makes the authors think it is suited to this task? Is there some feature of the problem that means this is an appropriate method? Ditto the novelty metric. Is is clear how to compute the metric, but there is a lack of argument or intution on why this is an appropriate way to measure the distance between two sets of polytopes.

The benchmarks used in the testing are sensible. It is always easy to raise questions about whether testing could be on a larger set of problems, but those presented here seem suitable for a conference paper. The competing methods evaluated against are appropriate and fair. The setting does not seem biased towards any of the methods.

The novel approach appears significantly to beat other learning approaches on two of the problem types. The new approach seems about the same as the other approaches on the facility location problems. It would be interesting to understand what it is about this problem that gives different results.

In table 1, wins are defined by number of nodes visited. In table 2, the time taken is used instead. Either method could be justified, but to change between experiments without good justification looks dubious.

A side benefit of the paper is that it results in determining branching rules which seem to perform well while not (intentionally) imitating strong branching. There is some investigation of why this is and why the technique works. Further research could build on this and the result may lead to further work on manually constructed branching rules.

There is a discussion section in the appendices. This seems very odd.

The paper is very well written - the language is easy to understand and the arguments being made are clear.

---

> ### Author Response · Authors · 2020-11-18
> **Response**
>
> Dear reviewer, thanks for reading our work and giving your valuable suggestions.
>
> (1) The RL method performs better on set covering and maximum independent set than on capacitated facility location. I think the reason is the former two are pure integer programming and the latter one is mixed-integer programming.
>
> (2) We used two metrics in table 1 and table 2 as we want to examine both the decision quality as well as the ability to solve challenging problems in a real setting. The 'clean' setting switches off other SCIP components, such as estimate node selection and cutting plane. This way, the evaluation eliminates the interference from other components of the solver to variable selection policy. Under the 'clean' setting, the solving nodes reflect the decision quality of variable selection policies only. So, we compare the decision quality of different methods under the 'clean' setting. The 'default' setting of SCIP will turn on all components inside SCIP, such as estimate node selection and cutting plane, which is tuned for solving real problems. So, We compare the ability to solve challenging problems of different methods under the 'default' setting.
>
> (3) For the discussion section, we will move it to the main text in the final version.

---

### Public Comment · ~Yunhao_Tang1 · 2020-11-10
**Questions on experiment details**

Many thanks to the authors for such a great work. I have a few questions regarding details of the experiments:

There seems to be a clear discrepancy between experiment results shown in Table 2 and those shown in Table 2 of [Gasse et al, 2019]. The performance of GCN seems to differ quite a lot - the results in your paper in Table 2 are worse by an order of magnitude compared to those in [Gasse et al, 2019]. As an example, consider the medium sized set covering problem: GCN requires 12k nodes as shown in Table 2, while in [Gasse et al, 2019] it takes about 1.4k on average. It also seems that similar discrepancies can be consistently found for other problems and for both medium/hard instances.

I am assuming that the authors build on the code provided by [Gasse et al, 2019] from the code link below. Though I understand that exactly replicating the results is not technically possible, but I am fairly surprised that the results are different by an order of magnitude.  Because the major baseline comparison is to contrast RL with imitation learning - as a result, the performance of imitation learning algorithm is of critical importance.

I really appreciate it if you could specify the potential setup differences that might lead to such a discrepancy in the performance of GCN.

=== reference ===
Gasse et al, 2019: Exact combinatorial optimization with graph convolutional neural network
code: https://github.com/ds4dm/learn2branch

---

### Author Response · Authors · 2020-11-17
**Questions about the experiment results.**

(1)  We focus on a more realistic industrial setting, and the data sets are not identical to Gasse et al. In many realistic industrial settings, there is a graph backbone behind the family of graph problem instances, and the family is generated by modifying the backbone. We follow this setting and generate our instances from a backbone. For example, in the maximum independent set, we generate a backbone graph with 350 nodes. All the other graphs for training, validating, testing is generated upon the backbone.

(2)The metrics used are different. Gasse et al. use the 1-shifted geometric mean of nodes while we report the average of the nodes. Also, Gasse et al. report the mean nodes for instances solved by all methods while we reported the average nodes for all instances. In some sense, averaging on all instances is unfair, so we will update the statistics in Table 1 and Table 2.

(3)Another difference is we use training data with a smaller size in the maximum independent set and capacitated facility location problems. As a result, the gap between training data and transfer data becomes larger and this factor impairs the generalization of GCN. SVM is less sensitive to this size change, so it performs quite well in our experiments.

---

### Decision · Program_Chairs · 2021-01-07
**Final Decision**

**Decision:**

Reject

**Comment:**

The paper describes an RL technique to learn how to branch in discrete optimization.  This advances the state of the art in comparison to previous imitation learning techniques.  However, the reviewers and a public reader raised concerns about the validity of the experiments due to several inconsistencies and differences with previous work that might suggest some cherry picking.  This is too bad since the reviewers really liked the work, but it is important to make sure that the experimental evaluation is done fairly.  I read the paper and I share the concerns regarding the experimental methodology.  Hence the experimental evaluation needs to be revised before publication.